

# Environmental Characteristics Associated with the Tropical Transition of Mediterranean Cyclones

Lisa Bernini[1,2,5], Leone Cavicchia[3], Fabien Desbiolles[4,5], Enrico Scoccimarro[3], and Claudia Pasquero[5]

[1]University of Genoa, Genoa, Italy
[2]CIMA Research Foundation, Savona, Italy
[3]Euro-Mediterranean Centre on Climate Change (CMCC), Bologna, Italy
[4]Laboratory of Space Geophysical and Oceanographic Studies (LEGOS), Toulouse, France
[5]University of Milan - Bicocca, Milan, Italy

**Correspondence:** Lisa Bernini (lisa.bernini@cimafoundation.org)

**Abstract.** Cyclonic perturbations in the Mediterranean region sometimes acquire characteristics typical of tropical cyclones, such as a deep inner warm core. In these cases, they become very intense structures that can cause large precipitations and significant damage. In this study, the environmental conditions during the intensification of cyclones are investigated using reanalysis data. A comparison of the conditions associated with the evolution of classical and intense cold-core extratropical

cyclones and those associated with the development of tropical-like disturbances highlights the characteristic that favors the conversion: a much larger potential intensity and a weaker vertical wind shear. The larger potential intensity associated with Mediterranean tropical-like cyclones comes from both higher SST and a strong PV-intrusion that destabilizes the air column. Sea surface cooling induced by the cyclones is further shown to play a role in the dissipation of tropical-like cyclones. Future research should focus on the role of potential intensity as a precursor for Mediterranean tropical-like cyclone forecasting,

improving predictive capabilities and risk mitigation strategies in the Mediterranean region.

## 1  Introduction

Various weather-induced natural hazards in the Mediterranean region like heavy rainfall, floods, and windstorms, are associated with Mediterranean cyclones (Lionello et al., 2006; Flaounas et al., 2022). These cyclones are extra-tropical cyclones (ETC) that have baroclinic origins, related to the deviation of the jet stream meanders over the Mediterranean Sea (Flocas, 2000; Fita

et al., 2007; Flaounas et al., 2015). Orography also plays a role in this region surrounded by mountains and makes some areas are more prone to cyclogenesis than others (Buzzi et al., 2003; Campins et al., 2011; McTaggart-Cowan et al., 2010b).

In rare cases, some cyclones develop in their mature stage characteristics similar to those of tropical cyclones (TC): an axisymmetric, deep warm core, generally with a windless cloud-free center surrounded by strong winds (Fita et al., 2007; Tous et al., 2013). Those Mediterranean Tropical-Like Cyclones (MTLC), also known as Medicanes - a portmanteau of

**Medi**terranean Hurri**canes** - are particularly severe: their peak strength can reach category 1 of the TC Saffir-Simpson scale (Akhtar et al., 2014).



They have recurrently affected, among others, Libya, Italy, and Greece, inflicting loss of human lives, environmental damages, and billions of Euros in losses (Bakkensen, 2017; Nastos et al., 2018). In addition, climatological studies suggest that in a warming climate, their intensity and duration may increase (Cavicchia et al., 2014b; González-Alemán et al., 2019). Hence
the necessity to properly understand their development and intensification processes.

There is no consensus in the scientific literature about the precise definition of MTLC, a reason that has recently led the scientific community to put effort into focusing on key characteristics and processes associated with Medicanes (Miglietta et al., 2025). Despite having in their mature stage some common characteristics with TC, their cyclogenesis is different. They develop over much colder temperatures than what can be found in the Tropics (Tous et al., 2013; McTaggart-Cowan et al.,
2010a) and they originate from baroclinic eddies. Indeed, they first develop as typical Mediterranean extratropical cyclones, meaning through the intrusion of a Potential Vorticity (PV) streamer into the Mediterranean (Raveh-Rubin and Flaounas, 2017; Flaounas et al., 2021).

After that initial phase, the development of the MTLC can follow different intensification processes. Similar to TC, it can be associated to the wind-induced surface heat exchange meschanism (WISHE, Emanuel (1986)). It means that MTLC
intensification is due to the positive feedback between the sea surface heat fluxes and the surface wind (Emanuel, 2005; Miglietta and Rotunno, 2019). In those cases, the large release of latent heat favors the development of a warm core, as in TC, and deep convection is significant to maintain the system.

Another mechanism that leads to the development of a warm core is associated with the enhancement of the low-level circulation by the upper-level PV anomaly, which in turn increases the sea-surface fluxes (Fita and Flaounas, 2018). In those
cases, the warm core is mainly a consequence of warm air seclusion in the cyclone's inner core by surrounding colder air (Mazza et al., 2017; Miglietta and Rotunno, 2019), and convection becomes rather weak during the mature stage of the cyclone.

The occurrence of MTLC is rare, with an annual frequency depending on the definition considered but on the order of very few cases per year (Tous and Romero, 2013; Ragone et al., 2018; Zhang et al., 2021; de la Vara et al., 2021). For this reason, the mechanisms responsible for their development have typically been analyzed in a relatively limited number of real case studies
(Varlas et al., 2023; Tous and Romero, 2013; Miglietta and Rotunno, 2019; Gutiérrez-Fernández et al., 2024). A climatological study based on downscaled reanalysis data focused on the environmental conditions during which they develop (Cavicchia et al., 2014a), highlighting the presence, during their initial phase, of low wind shear, upper-level cold intrusions, high moisture content, and strong vorticity. Those conditions had been quantified as anomalies with respect to the climatological seasonal cycle. Those, however, are conditions that to some degree characterize the environment of cyclonic perturbations in general.
It remains thus to be investigated whether there is any peculiar characterization of the conditions that favor the transition of classical extratropical cyclones into warm core cyclones. In this study, the evolution of the environmental characteristics alongside the Mediterranean cyclones' lifetime has been analyzed to identify environmental precursors that could explain why some Mediterranean cyclones develop Tropical characteristics.



## 2 Materials & Methods

### 2.1 Cyclone tracks and atmospheric conditions

Mediterranean cyclone tracks are taken from a new dataset provided by Flaounas et al. (2023). It consists of composite cyclone trajectories and intensities detected by ten different cyclone detection and tracking methods applied to hourly data of the ERA5 reanalysis (Hersbach et al., 2020) in the 42 years of 1979-2020. In the following work, we retained the cyclones tracked by at least five of the ten tracking algorithms (confidence level of five). Such a confidence level is a trade-off between "robustness" and "completeness" of the final dataset and has already been used in other studies (Givon et al., 2023). The original cyclones of Flaounas et al. (2023) have been tracked on the Extended Mediterranean Sea Region: -20°E;45°E;20°N;50°N. To focus only on Mediterranean Cyclones, Atlantic Cyclones have been discarded by reducing the domain to: 6°W;45°E;30°N;50°N. The cyclones that stay less than six hours over the Mediterranean Sea have also been discarded to not take into account the thermal lows present over the Sahara desert.

The maximum intensity of the cyclone is defined as the lifetime minimum Sea Level Pressure, SLP, at the cyclone center, and the time of its first occurrence is defined as time 0. Data are then organized into an intensification phase (before time 0) and a weakening phase (after time 0), despite in a few cases short re-intensification periods might occur.

Based on the position of the tracked cyclones, 3D fields of temperature, moisture, wind, rainfall, geopotential height, heat fluxes, and potential vorticity have been extracted from the ERA5 reanalysis dataset in an area of $10° \times 10°$ surrounding the center of the depression. Anomalies have been computed as departures from the climatological seasonal cycle computed as a 7-day running mean of the daily mean values over the 42 years.

### 2.2 Hart cyclones phase space diagram

The cyclone phase space (CPS) diagram (Hart, 2003) has been applied to distinguish between typical asymmetric ETC with a cold inner core and axisymmetrical TLC with a deep inner warm core. To construct this CPS, three parameters are needed:

- $B$, the thermal asymmetry of the cyclone in the lower troposphere.

- $V_l$, the lower tropospheric thermal wind

- $V_u$, the upper tropospheric thermal wind

Those parameters are computed using the geopotential height field Z at three different vertical levels (900 hPA, 600 hPA, and 200 hPA) within a circle of radius 100 km around the cyclone center. They are mathematically defined as follows:

$$B = \overline{Z_{600}(\mathbf{x},t) - Z_{900}(\mathbf{x},t)}\Big|_R - \overline{Z_{600}(\mathbf{x},t) - Z_{900}(\mathbf{x},t)}\Big|_L \qquad (1)$$



where the over-bar indicates the mean over the area of a semicircle of radius 100 km, located to the right (subscript $R$) or to the left (subscript $L$) of the storm trajectory.

$$-|V_l| = \left.\frac{\partial(\Delta Z)}{\partial \ln(p)}\right|_{900}^{600} \tag{2}$$

$$-|V_u| = \left.\frac{\partial(\Delta Z)}{\partial \ln(p)}\right|_{600}^{200} \tag{3}$$

with $\Delta Z = max(Z) - min(Z)$ (at the same pressure level, in a region of radius 125 km around the cyclone center).

In Hart's view, a TLC is a non-frontal system characterized by thermal symmetry, while an ETC is a frontal system that is thermally asymmetric. Mature TC have values of $B$ of approximately zero while developing ETC have large positive values of $B$. The symmetry condition for distinguishing a TLC from a ETC is conventionally set at $|B| < 10$ m.

The parameters $V_l$ and $V_u$ instead are related to the radial gradient of temperature in the cyclone. In a cold-core structure, the geopotential height perturbation increases with height. Cold-core structures like ETC have negative values for both $-V_l$ and $-V_u$. Conversely, in a warm-core structure, the geopotential height perturbation is larger closer to the surface than at higher levels. Warm-core structures like TC have positive values for both $-V_l$ and $-V_u$. Hybrid and transitioning cyclones may have a sign of $-V_l$ that is different from $-V_u$.

In the present study, following the work of Cavicchia et al. (2014a); Gaertner et al. (2018); Picornell et al. (2014); Walsh et al. (2014), we define MTLC as cyclones that during part of their lifetime develop a deep warm core: both $-V_l$ and $-V_u$ must be positive for at least six hours while they are over the sea. The ETC are the ones with a cold core, i.e. they have negative values for both $-V_l$ and $-V_u$, for the whole lifetime or they have a short duration positive value for $-V_u$ only (less than six hours). The remaining cyclones are considered as hybrid and are not included in the study.

In the dataset we used, adding the B parameter threshold criterium to discriminate MTLC from ETC decreases the number of warm core cyclones by 5%. This is likely due to the fact that most tracking algorithms used in Flaounas et al. (2023) are already considering symmetry criteria to identify a structure as a cyclone. Therefore, in this work we only use core temperature ($-V_l$ and $-V_u$ parameters) to discriminate between ETC and MTLC.

## 2.3 Potential Intensity

Potential Intensity (PI) represents the theoretical maximum intensity a tropical cyclone (TC) can attain under specific environmental conditions. It is determined by different thermodynamic factors such as sea surface temperature, atmospheric temperature in the air column, and moisture availability. Following many influential studies (Vecchi and Soden, 2007; Garner et al., 2009; Kossin and Camargo, 2009; Camargo et al., 2013; Ramsay, 2013; Vecchi et al., 2013; Wing et al., 2015; Emanuel, 2018; Shields et al., 2020; Xu et al., 2019), we adopt the computation of PI based on Bister and Emanuel (2002):

$$PI^2 = \frac{C_k}{C_d}\frac{T_s}{T_o}(CAPE^* - CAPE_{env}) \tag{4}$$




where $C_k$ and $C_d$ are the surface exchange coefficients for enthalpy and momentum; $T_s$ and $T_o$ are the sea surface and outflow

temperatures; $CAPE^*$ is the convective available potential energy of an air parcel lifted from saturation conditions at the sea

surface temperature (referred to as *hurricane CAPE* in the following), and $CAPE_{env}$ is the convective available potential

energy of a near surface air parcel (referred to as *environmental CAPE* in the following).

    The potential intensity was computed using the Python packages described in Gilford (2021). The function takes as input

ERA5 sea surface temperature (SST), and temperature and relative humidity profiles at the following pressure levels: 2m,

900hPa, 850hPa, 700hPa, 500hPa, 400hPa, 300hPa, 200hPa, 100hPa, and 50hPa. The ratio $\frac{C_k}{C_d}$ was set to 0.9.

    In computing the SST and PI composites over the previously mentioned $10° \times 10°$ areas, grid cells that were over land have

been excluded. Then, we also performed a sensitivity test by assigning $PI = 0 \ m.s^{-1}$ in grid points over land areas. While

this adjustment significantly reduces the box-averaged PI values, the results of the analysis remain qualitatively unchanged,

indicating that the general conclusions of the work are not sensitive to the way in which land points are treated.

## 3 Results

### 3.1 Classification of cyclones

Considering all the criteria previously mentioned, over the 42 years studied, the total number of cyclones present in the dataset is

2026. Of those 2026 cyclones, an average of 23 cyclones per year are classified as ETC cyclones (959 cyclones in total) and 3.4

cyclones per year are classified as MTLC cyclones (142 cyclones in total). The rest are intermediate or hybrid cyclones, which

were not considered in the following study. The typical ETC have been divided into two groups based on their intensity: the

20% cyclones with the lower Sea Level Pressure (SLP) value have been classified as "Intense ETC". There are 192 cyclones,

a headcount comparable to the size of the MTLC group. The remaining 767 cyclones are thereafter labeled as "ETC", and

referred to as normal cyclones.

It should be noted that the number of Medicanes typically reported in the literature is about 1.5 per year (Romero and

Emanuel, 2013; Cavicchia et al., 2014b; Ragone et al., 2018; Zhang et al., 2021), considerably smaller than the 3.4 per year

that we find. However, in several cases, the number of occurrences has been obtained by explicitly tuning selection criteria

used for the definition of Medicanes to obtain an average of 1.5 events per year, a number that originated from an informal

register that was maintained by the Meteorological group at the University of Balearic Islands (see www.uib.es/depart/dfs/

meteorologia/METEOROLOGIA/MEDICANES). By varying a threshold parameter within a reasonable range, Cavicchia et al.

(2014b) found that the number of events varied by a factor of three. Similar sensitivity was presented in Ragone et al. (2018).

For those reasons, we choose here to retain all warm core cyclones that have been obtained using the criteria defined above for

subsequent analysis of MTLC.

    The evolution of the number of MTLC and intense ETC in the period from 36 h before to 36 h after the cyclones' maximum

intensity is shown in **fig. 1**, with the count decreasing both for negative and positive time intervals due to the limited duration of

the tracks. On average, the tracks last 64 hours for normal and intense ETC, while they last much longer (89 hours) for MTLC,

the difference between the classes being statistically significant at the 99.9% confidence level.





**Figure 1** also presents the proportion of MTLC that have a full warm core (positive values for both $-V_l$ and $-V_u$) at any given time lag from 0. About one-third of MTLC have a full warm core at the time of peak intensity, developed in most cases less than 6 hours before, and a total of 75% have at least a lower or an upper warm core at time t=0, while the remaining MTLC develop a warm core in the subsequent hours. At 36 hours before the maximum intensity, none of the cyclones in the MTLC category present a warm core. It means that they are still in their typical cold core or hybrid conditions at those time-steps. For this reason, the difference in the tropospheric conditions between ETC and MTLC at 36 hr before the peak intensity is investigated in section **3.3** to identify possible precursors to the tropical-like transition.

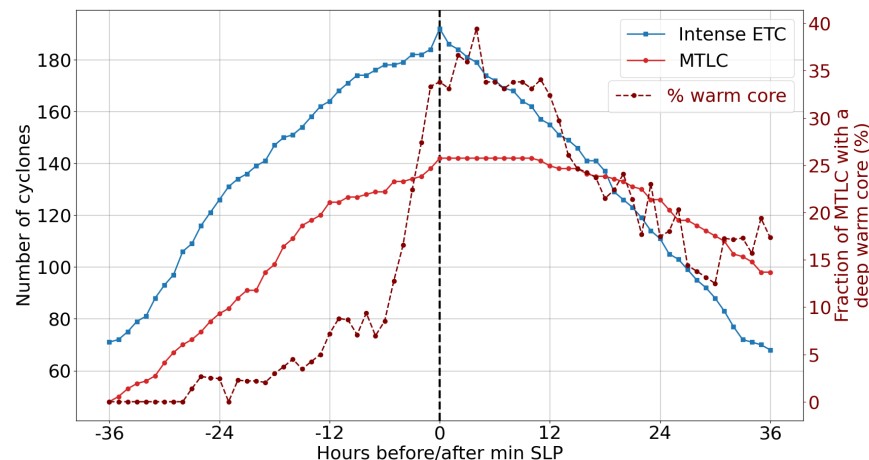

**Figure 1.** Time evolution with respect to minimum SLP of the number of intense ETC (blue solid line) and MTLC (red solid line), and proportion of MTLC that have a warm-core at each time step (red dotted line).

## 3.2 General characteristics

Cyclones in the Mediterranean are known to occur preferentially in specific regions, including the Gulf of Genoa, the Adriatic Sea, the Black Sea, close to Cyprus, and in northwest Africa, at the leeward side of the Atlas Mountains (Thorncroft and Flocas, 1997; Maheras et al., 2001; Flocas et al., 2010; Campins et al., 2011; Ulbrich et al., 2012; Reale and Lionello, 2013; Ammar et al., 2014; Maslova et al., 2020). As revealed in **Fig. 2**, which shows the spatial density of tracks separated into ETC and MTLC, the overall distribution of the cyclones analyzed in this study matches the geographical localization of Mediterranean cyclones, but cyclones of different categories are preferentially located in different positions. The most intense ETC are mainly located in the Gulf of Genoa while cyclones that develop a warm core are mainly present in the western Mediterranean Sea and the Ionian Sea, with some present in the Black Sea, in line with previous studies (Cavicchia et al., 2014b; Tous and Romero, 2013). It is important to note that at 36 hours before their maximum intensity, the timestep at which we investigate the environmental conditions associated with each category of cyclone in the next section, only one third of intense ETC have



their center over the sea, while about half of MTLC have their center over the sea. This difference is consistent with the already mentioned role played by the air-sea exchanges in the development of MTLC.

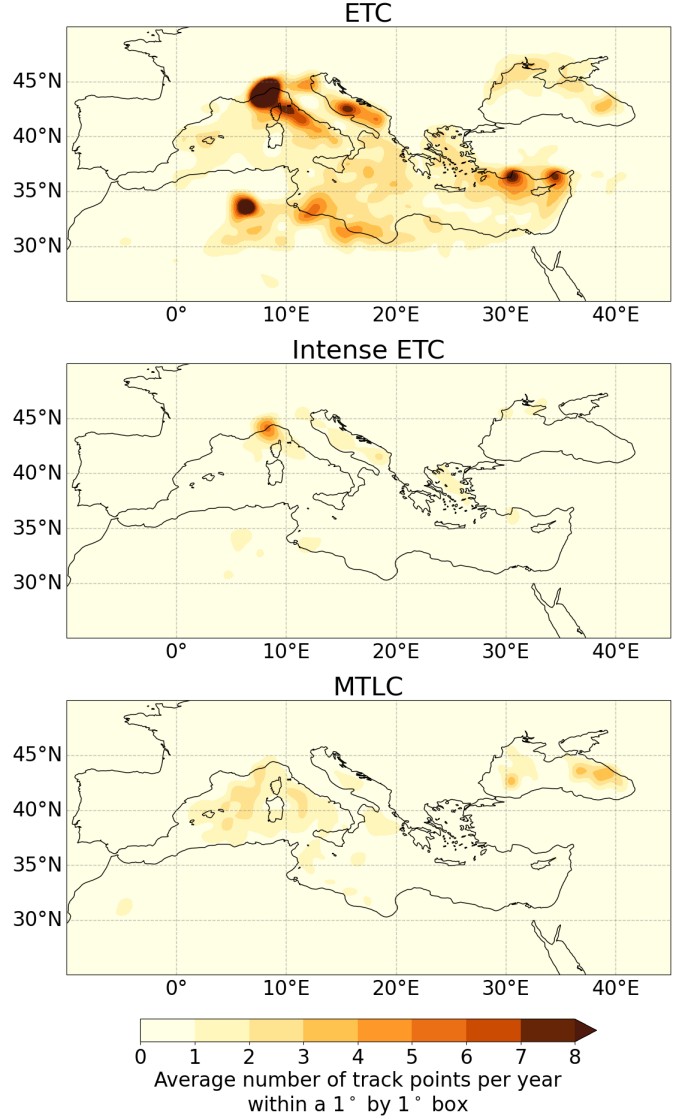

**Figure 2.** Spatial density of cyclone tracks expressed as the average count of track points per year within a $1° \times 1°$ box.

The seasonality of Mediterranean cyclones indicates a preferential occurrence in Spring and Fall (Campins et al., 2011). This is reflected both in the occurrence of ETC and MTLC (see **fig. 3**). Intense ETC occurrence presents a well-defined maximum in Spring. MTLC instead are most frequent in October, when the large SST favor air-sea heat exchanges and therefore promote the establishment of the warm core (Gaertner et al., 2018). However, several MTLC are also found in Spring. The different





seasonality in their occurrence is reflected in the composite sea surface temperature over which they develop, shown in **fig. 4** at 36 hr prior to peak intensity: SST is about $3^o$C warmer for MTLC than for intense ETC.

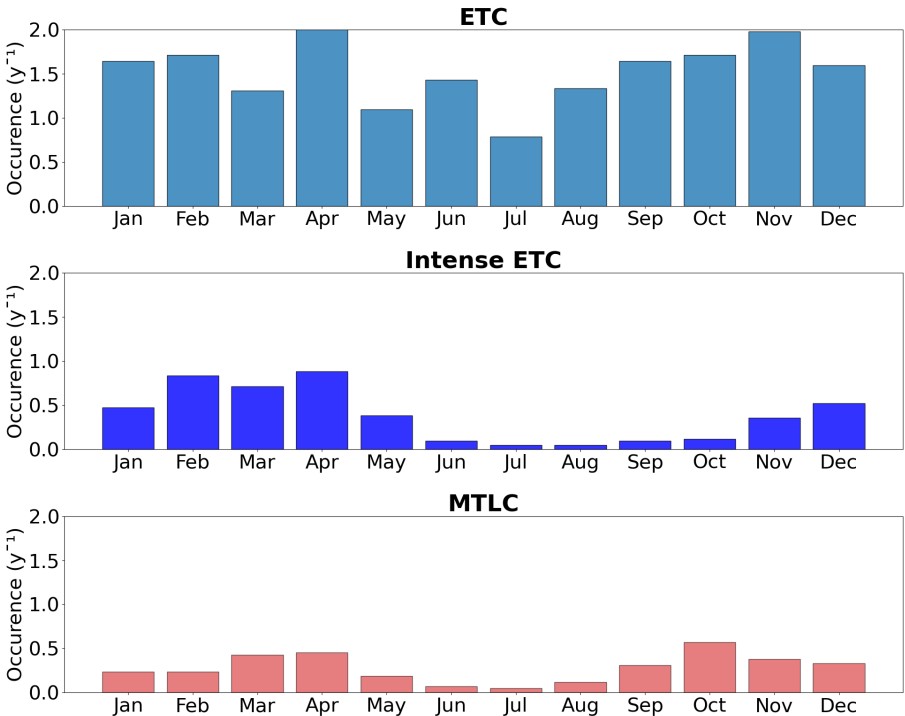

**Figure 3.** Seasonal cycle of ETC (light blue, top row), intense ETC (dark blue, middle row), and intense MTLC (red, bottom row) occurrence.





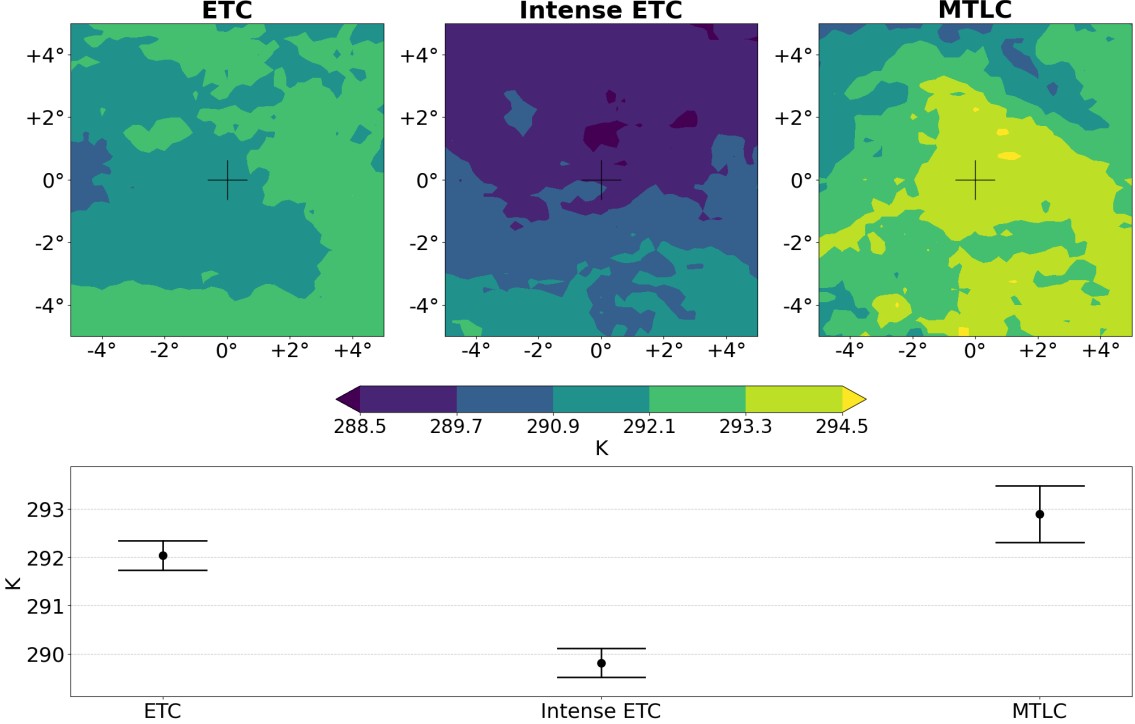

**Figure 4.** Sea Surface Temperature (SST) composites 36h before the time of minimum SLP.

ETC and MTLC are also characterized by different peak intensities: the largest SLP anomaly is on average -10 hPa for ETC and -17 hPa for MTLC, whereas intense ETC are on average even stronger than MTLC (in terms of their depression, which averages at -21 hPa). Consistently, peak winds are comparable between intense ETC and MTLC, and are much smaller in typical ETC (**fig. 5**).

Wind composites (**fig. 5**) have a maximum in the southwest quadrant. In MTLC, the maximum azimuthally averaged winds are found closer to the center than in ETC (see **sup. fig. A2**). However, large heterogeneities exist as the standard deviation of the radius of maximum winds is over 100km in all classes.

A major difference emerging between MTLC and intense ETC is in their associated precipitation: despite the similar cyclone intensity between the two categories, hourly rainfall is much higher in MTLC, where the azimuthal average peaks to values that are 40% larger than for intense ETC (**fig. 6** and **sup. fig. A3**). Following the literature on TC and the recent evidence that rain intensity is at least as important as wind velocity to explain the socioeconomic loss associated with cyclones (Bakkensen et al., 2018; Qin et al., 2020; Wen et al., 2018), those composites confirm that MTLC have the potential to cause more significant damages than their extratropical counterparts.

The precipitation at the peak intensity of ETC is localized in the North-East quadrant (**fig.6**). This spatial distribution is similar to the one reported in the literature (Field and Wood, 2007; Yettella and Kay, 2017; Naud et al., 2018) for typical extratropical cyclones and is due to the presence of the warm-frontal region. However, our dataset doesn't clearly show the



typical comma-shaped precipitation pattern. This is probably related to the fact that the composites have been computed using longitude-latitude coordinates rather than using the along track and the across track directions. Despite many cyclones move eastward carried by the westerly winds typical of mid-latitude conditions, several factors affect their translation velocity and thus influence the longitude/latitude composite.

The maximum precipitation for the warm-core cyclones is found in the North-West quadrant (**fig. 6**). Usually, the peak of rainfall in TC occurs at the forefront of the storm center, where the uneven circulation induced by friction in the moving storm generates low-level convergence, significantly intensifying deep convective activity (Lonfat et al., 2004; Du et al., 2023). This would indicate that most of the MTLC, or at least those with larger precipitation rates, move in the north-west direction. The wind roses, shown in **sup. fig. A1**, confirms this hypothesis.

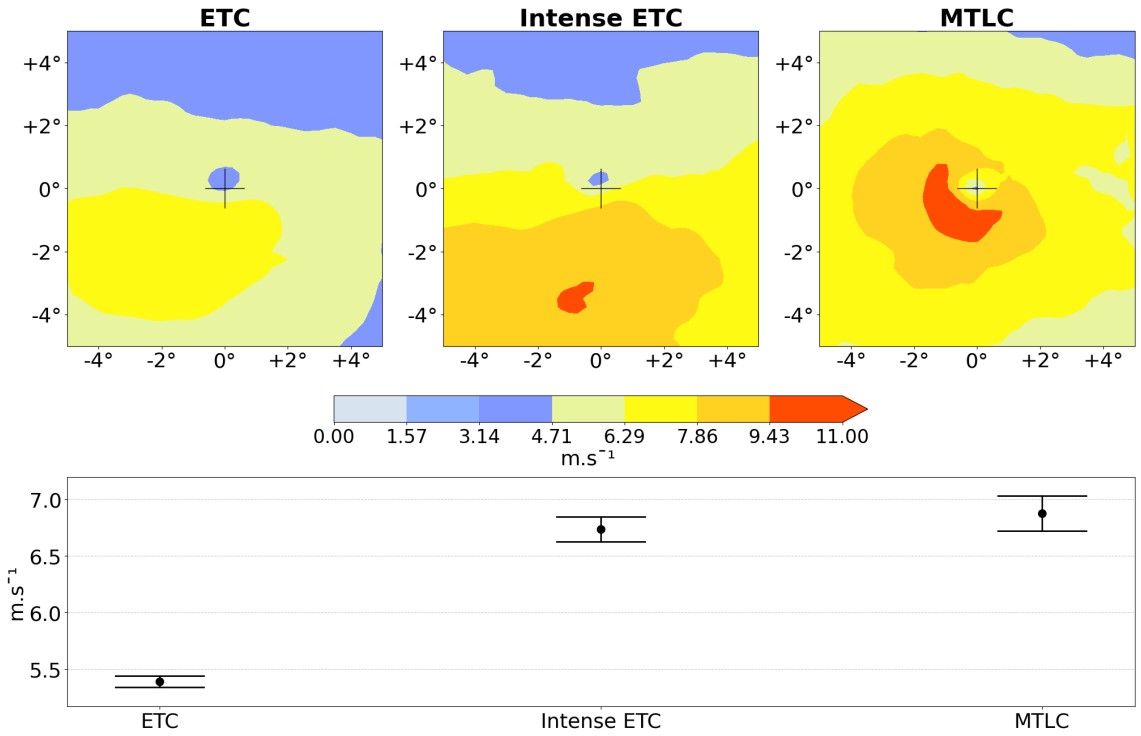

**Figure 5.** 2D composites centered on the cyclones of surface wind 36h before the time of minimum SLP.





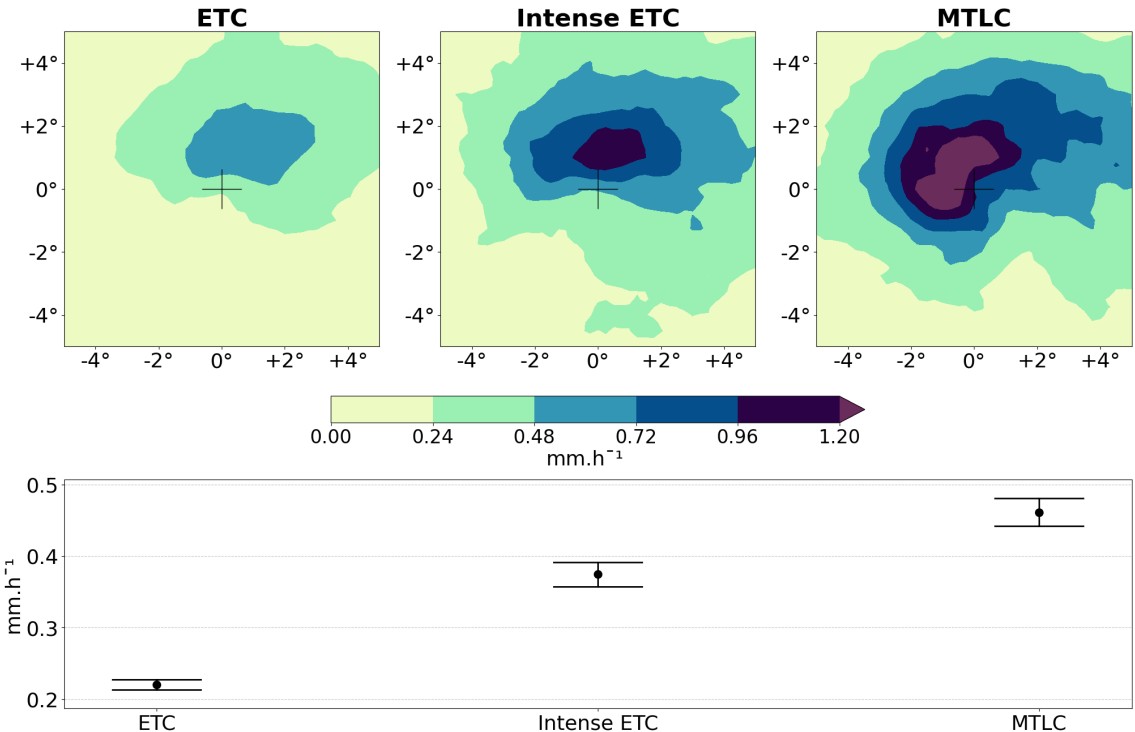

**Figure 6.** 2D composites centered on the cyclones of precipitation 36h before the time of minimum SLP.

## 3.3 Intensification phase

The composite evolution of surface wind speed in the different cyclone categories is shown in **Fig. 7**. In this case, wind speed has been averaged over a 10°x10° longitude-latitude box centered at the minimum sea level pressure location. While this metric also indicates similar peak intensities of MTLC and intense ETC, the intensity evolution is quite different, indicating a statistically significant larger intensification rate for MTLC in the earlier stages, triggered by larger air-sea (latent and sensible) heat fluxes (**Fig. 8**). Intense ETC, despite having much stronger surface winds than classical ETC, show quite similar surface

fluxes, suggesting that the wind-induced surface heat exchange (WISHE) feedback is not active in those cases, whereas it can drive the evolution of MTLC. To investigate possible reasons for the observed differences, we next analyze the conditions under which cyclones evolve, focusing on the characteristics 36 hr before their peak intensity, when none of them has developed a deep warm core.




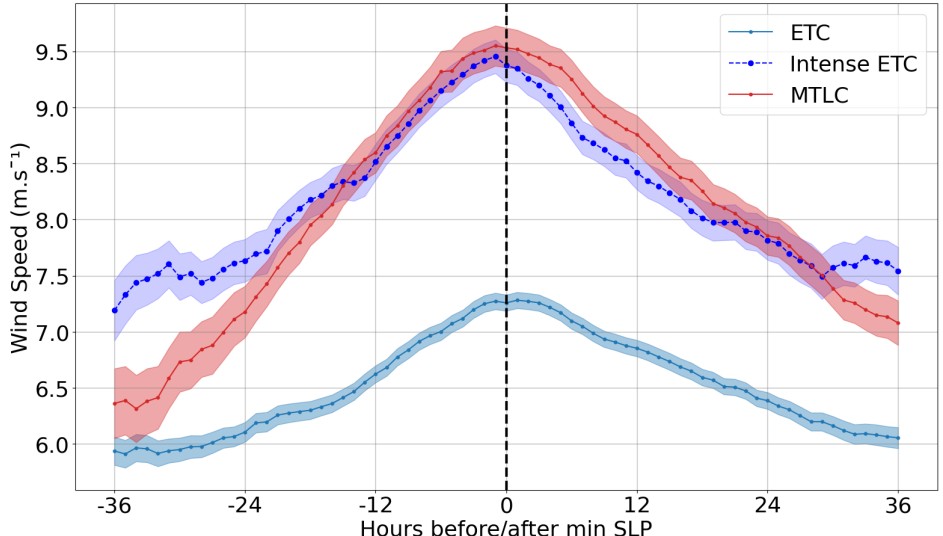

**Figure 7.** Composite time evolution of mean 10-m wind speed in a 10° by 10° box centered on the cyclone. Only the points over the sea are considered to compute the composite. The vertical black dashed line indicates the time of minimum sea level pressure. Shading around the solid line indicates the standard error of the mean.

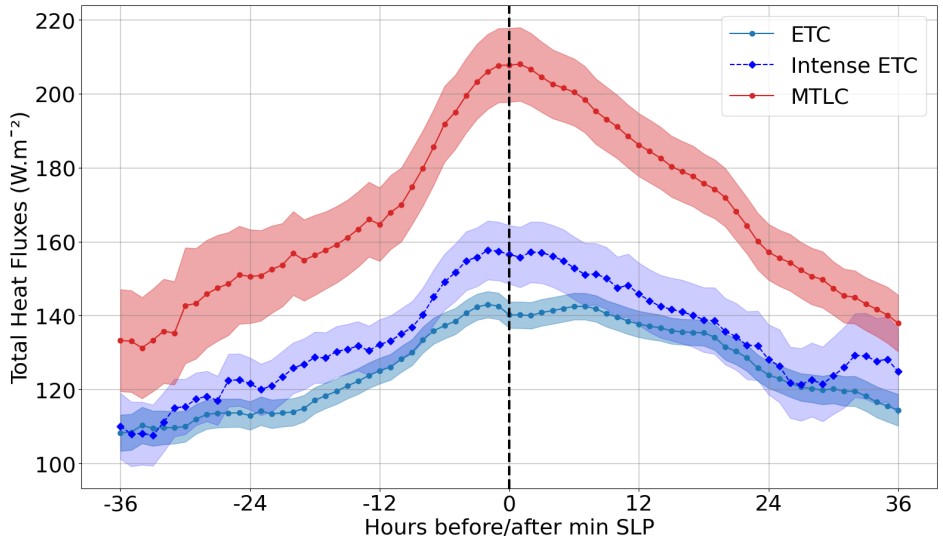

**Figure 8.** Composite time evolution of mean air-sea total (latent and sensible) heat flux in a 10° by 10° box centered on the cyclone. Only the points over the sea are considered to compute the composite. The vertical black dashed line indicates the time of minimum sea level pressure. Shading around the solid line indicates the standard error of the mean.

As already mentioned in the introduction, the origin of Mediterranean cyclones is the presence of a trough in the upper troposphere. The meandering of the jetstream advects positive potential vorticity (PV) anomaly into the region and is associated






with cyclone development (Dolores-Tesillos et al., 2022; Sanchez et al., 2023). It also favors upper-level divergence, upward motion, and the intensification of the surface depression. The positive upper-level PV anomaly is present in the analyzed cyclones, as shown in the 300hPa composite maps 36h before the cyclone peak intensity (**fig. 9**). The average PV anomaly is relatively weak for ETC, larger for intense ETC and even larger for MTLC, but the very large differences among cyclones of

the same class indicate that the PV anomaly strength is neither a sufficient nor necessary condition for MTLC development.

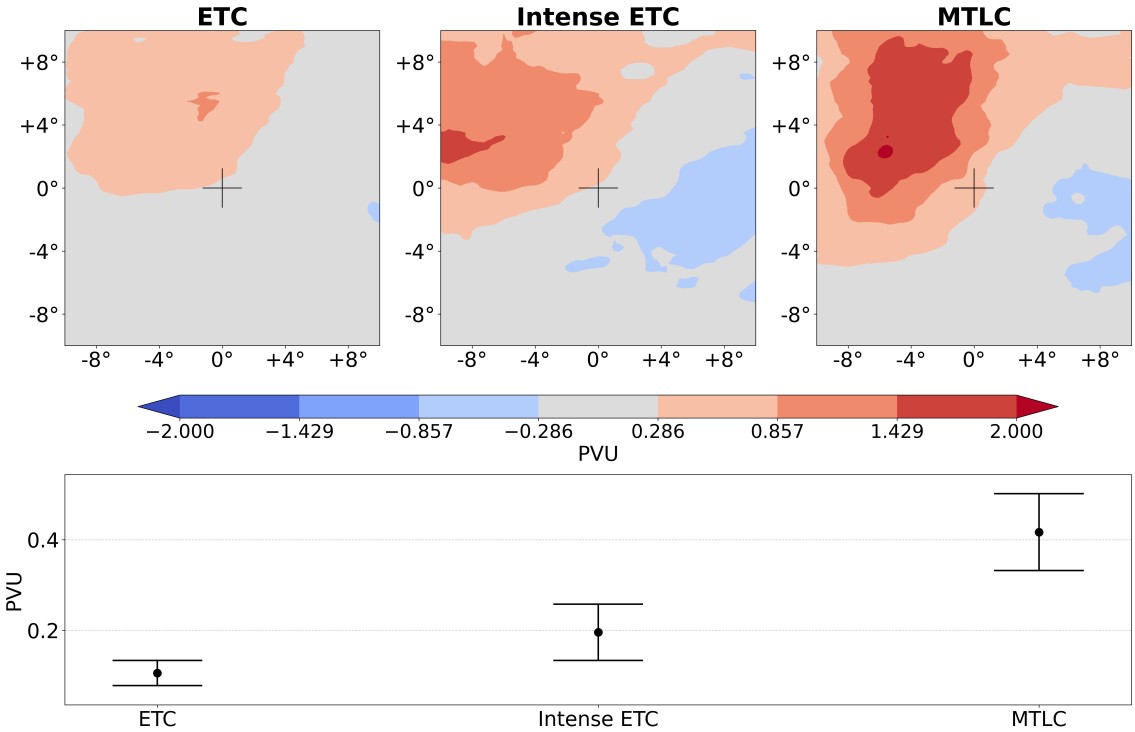

**Figure 9.** 300hPa Potential Vorticity (PV) anomaly with respect to climatological conditions, for the different cyclone classes. Top: 2D composites centered on the cyclones, 36h before the time of minimum SLP. Bottom: box average of the PV anomalies shown in the maps above; error bars indicate the standard error of the mean over the different cyclones in each class.

We next show composite maps of potential intensity 36 hr prior to peak intensity ( **fig. 10**). MTLC have PI values exceeding the threshold typically associated with tropical cyclone development ($> 35$ $m.s^{-1}$; Emanuel (2010)), while PI is much lower for ETC and intense ETC. The MTLC high PI region is present in proximity of the cyclone, indicating that the perturbation itself contributes to increasing the potential intensity in an environment overall less prone to TC development, as discussed in

Emanuel et al. (2024).





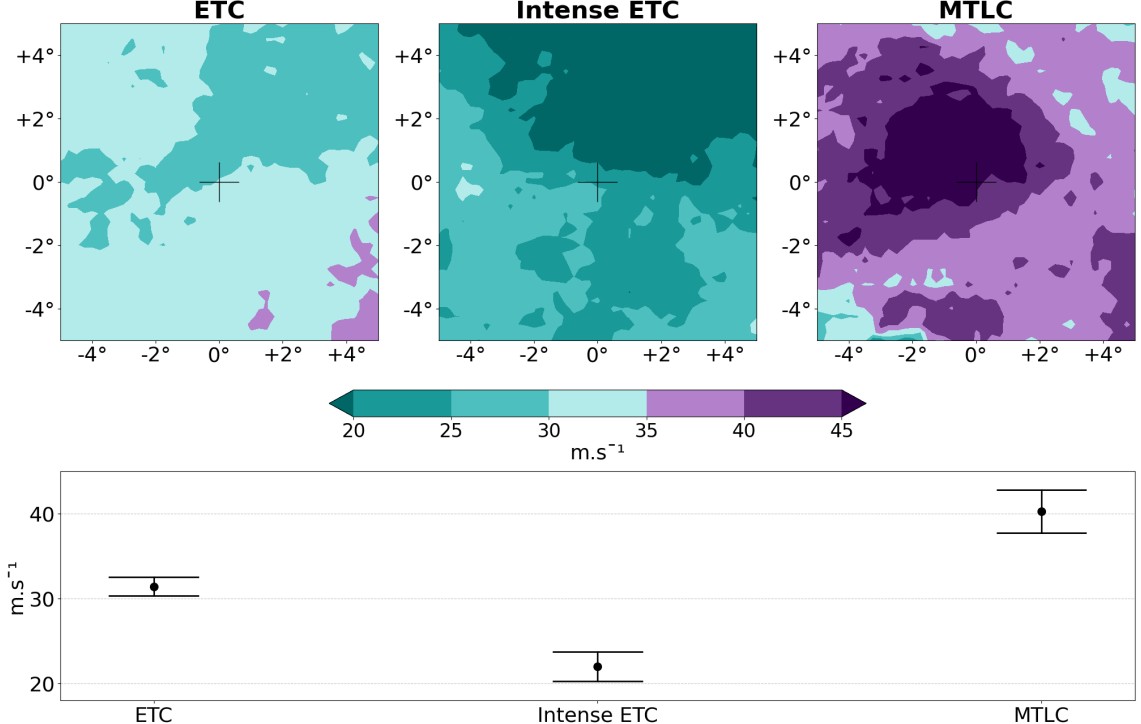

**Figure 10.** PI for the different cyclone classes. Top: 2D composites centered on the cyclones, 36h before the time of minimum SLP. The purple colors represent values of PI above $35\ m\ s^{-1}$, value below which tropical cyclones typically don't form (Emanuel, 2010). Bottom: box average of the PI shown in the maps above; error bars indicate the standard error of the mean over the different cyclones in each class.

To investigate what drives potential intensity differences, we compute PI from the climatological conditions at the cyclones' spatial and temporal coordinates. A stronger climatological PI is obtained for MTLC than for intense ETC (**fig. A5**). This difference (that has an average magnitude of $12\ m.s^{-1}$) is associated with their different seasonality, as MTLC occurrence peaks in fall (**fig. 1**), when the larger SST (**fig. 4**) are associated both to a larger magnitude of the temperature ratio term $\frac{SST}{T_o}$
and of the hurricane CAPE term in eq. (4). However, the climatological PI associated with MTLC presents less intense values than the actual PI obtained at their formation, indicating that there are some specific conditions when MTLC develop which increase the local PI value, resulting in a composite PI anomaly (actual minus climatological value) of up to $13\ m\ s^{-1}$ (**sup. fig. A6.a**), and with a mean of $6\ m\ s^{-1}$.

The main contribution to such an anomaly in PI for MTLC is due to the anomalous CAPE difference term in **eq. 4**, while
the $SST/T_o$ temperature ratio does not significantly differ from its climatological value (see **sup. fig. A6.b** and **A6.c**).

The CAPE difference term depends on sea surface temperature, on near-surface specific humidity (here taken at 2m asl), and on the temperature profile in the whole air column. To determine which variable has the most important influence on the anomaly produced in PI for MTLC, we compute the CAPE difference term inserted in the PI derivation using all climatological values except one.




For instance, while keeping climatological values for near surface specific humidity and tropospheric air temperature, we insert the actual values of SST and derive an anomalous hurricane CAPE that is then used in the PI computation. The difference between the obtained PI and the climatological PI is due to the effect of the SST anomaly only on hurricane CAPE, and is then composited onto the different MTLC and shown in **fig. 11a**.

The procedure is then repeated for anomalous near surface specific humidity (here taken at 2m asl, which affects the buoy-
ancy of the near surface air parcel), for anomalous near surface air temperature (which again affects the buoyancy of the near surface air parcel), and for the rest of the tropospheric temperature profile (which enters in the computation of parcel CAPE). All the different terms are shown in **fig. 11**.

They indicate that the negative anomaly in the mid-troposphere air temperature (see **fig. 12**) has the largest effect, followed by the positive SST anomalies. The anomalies in near-surface air properties have a minimal effect. The negative anomaly in
tropospheric temperature increases the density of the environmental air, contributing to the $CAPE^*$ increase. The positive SST anomaly increases the buoyancy of the saturated parcel used to compute hurricane CAPE. Together, the two anomalies significantly increase the hurricane CAPE and thus the PI with respect to climatological conditions. The linear superposition of the PI anomalies shown in **fig. 11** provides a total anomaly very close to the full one, indicating that nonlinear effects are small.

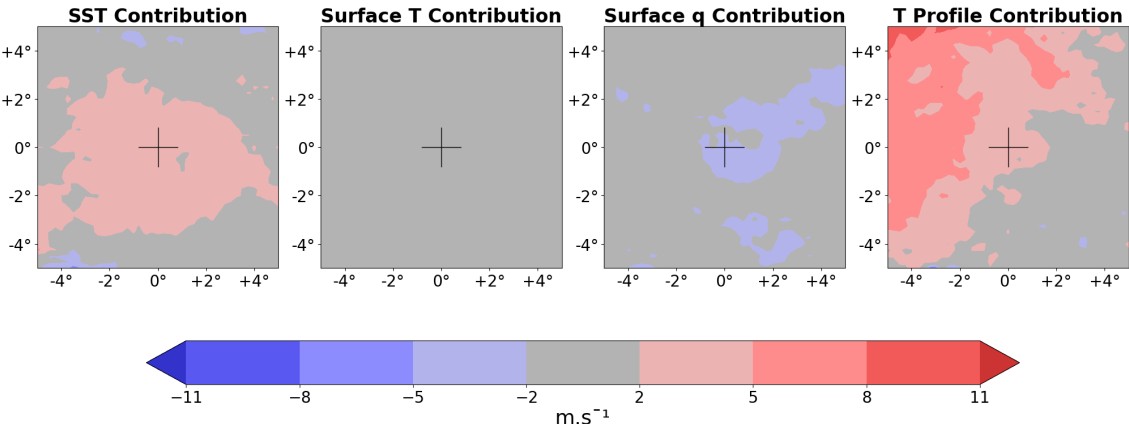

**Figure 11.** 2D composites of the contribution of different anomalous terms (as indicated in titles) to the PI anomaly with respect to climatology, through their effect on the CAPE difference term, for MTLC 36h before minimum SLP.





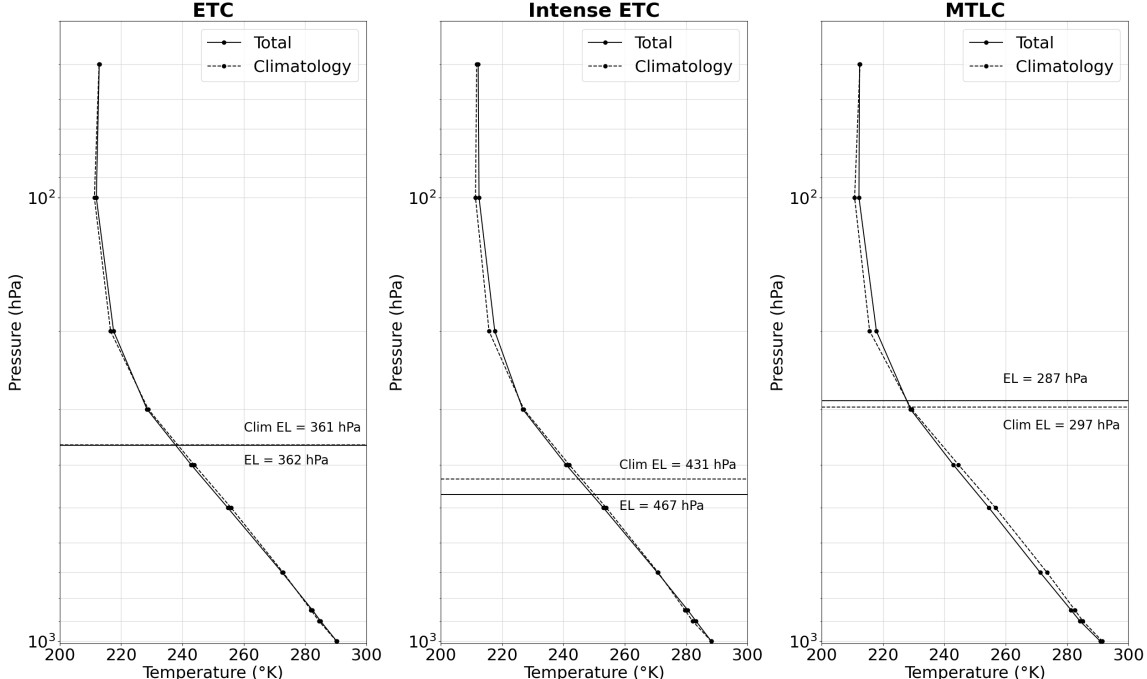

**Figure 12.** Temperature profiles 36h before the time of minimum SLP. The value is the mean temperature over a 5° by 5° box centered on the cyclone, but only the points over the sea are considered. The equilibrium level (EL) averaged over the same box is also indicated. For all variables, dashed lines correspond to climatological values and full lines indicate the actual values emerging from the composites.

In addition to PI, we also explore the vertical wind shear in the different categories of Mediterranean cyclones. Indeed, strong wind shear is known to have detrimental effects on the development of tropical cyclones through its ventilation effects (Kaplan and DeMaria, 2003; Elsberry and Jeffries, 1996; Emanuel et al., 2004; Wong and Chan, 2004), and its role on the formation of MTLC has been previously discussed (Tous and Romero, 2013; Cavicchia et al., 2014a). In our dataset, MTLC do present a weaker wind shear than intense ETC (**Fig. 13**), largely because of differences in the upper-level winds. At time t=-36hr, 300

hPa winds are stronger than in the climatology for all categories of cyclones, in line with their association with the meandering of the jet stream. However, they are less strong for MTLC and normal ETC than for intense ETC.





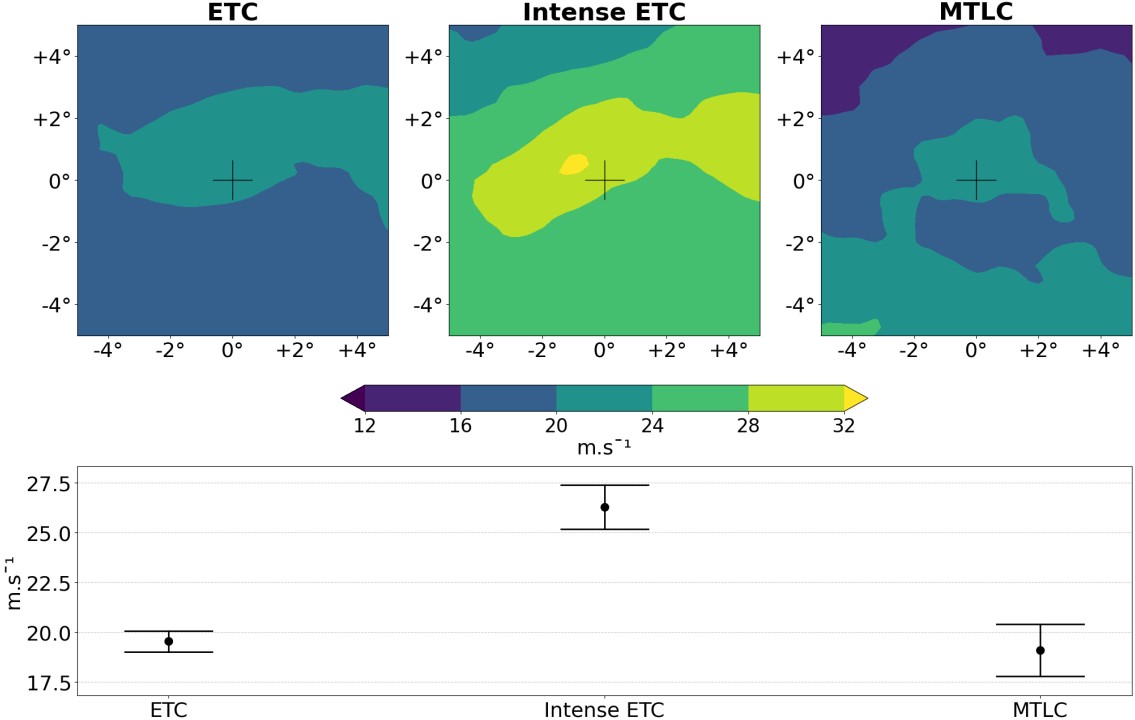

**Figure 13.** Wind shear composites 36h before the time of minimum SLP, for the different classes of cyclones. Here, the wind shear is defined as the difference between the wind at 300hPa and the wind at 850hPa.

In summary, we have shown above that in early stages MTLC have a stronger PI than ETC and develop in an environment with limited wind shear. Both circumstances favor the development of tropical-like characteristics onto the cyclonic structure generated by the intrusion of upper-level potential vorticity anomaly.

### 3.4 Potential cause of the MTLC decay

Landfall is usually cited as an important cause of deprivation of moisture supply in tropical cyclones, causing their dissipation (Anthes, 2016; Kaplan and DeMaria, 1995). However, in the present dataset, more than 75% of MTLC remain over the sea for 36hr after the peak intensity, while they significantly weaken **fig. 14** and in many cases loose their warm core (**sup. fig. A8**). Thus, other processes could be responsible for the onset of the dissipation phase.

In **fig. 14** the time evolution of MTLC potential intensity is shown between -36 hr and + 36 hr from time of peak intensity. A progressive decrease of the PI with time can be noticed starting already in early stages and certainly before peak intensity.



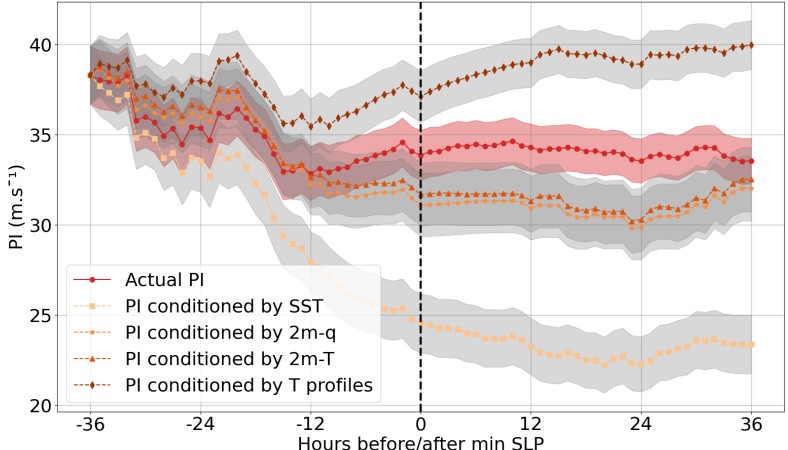

**Figure 14.** Composite time evolution of the mean PI for MTLC with respect to their minimum SLP (black dashed line). The red solid line represents the actual mean PI. Other lines refer to PI computed letting only one variable evolve in time and all the others fixed at their initial value: SST (square markers), 2m-temperature (triangle markers), 2m-specific humidity (star markers), tropospheric temperature profile (diamond markers). The mean has been computed in a 4° by 4° box centered on each cyclone. Shading around the solid line indicates the standard error of the mean.

We notice that at the same time there is a considerable reduction in SST (**sup. fig. A4**). The reduction in SST along the trajectory of MTLC could be driven by the action of the strong cyclonic winds, which are well known to generate cold wakes on the tracks of TC (Mei and Pasquero, 2013). We also notice that the translation speed of MTLC (and of intense ETC as well)

significantly drops starting 12 hr before peak intensity (**sup. fig. A9**). While the investigation of the reasons behind this change is beyond the scope of this work, we note that a slower translation speed both causes a colder anomaly at the sea surface (Mei et al., 2012) and leads to the cyclone spending more time over cool water. Those mechanisms generate a reduction in the air-sea fluxes because of the reduced thermodynamical disequilibrium between the upper ocean and the lower atmosphere associated with colder SST and are known to reduce the intensification rate of tropical cyclones (Mei et al., 2012).

The reduction in SST leads to a decrease in hurricane CAPE, ultimately resulting in lower PI. To investigate this influence, we computed additional time evolutions of PI by holding all variables constant at their initial values (either at -36hr, or at the first available track time for cyclones with a later track onset), except for one: SST, near-surface specific humidity, near-surface temperature, or the full atmospheric temperature profile (**fig. 14**). Among these, the PI computed with varying SST and fixed other variables shows the largest decrease, suggesting that the SST drop is the primary contributor to the PI reduction.

In this regard, we notice that the composite SST reduction of 3K associated with the MTLC is a very strong anomaly, compared to cold wakes of TC (Mei and Pasquero, 2013). This difference, and hence the stronger effect of the cold wake feedback on the intensity of the cyclone, can be related to the shallow mixed layer present in the Mediterranean sea (typically with a depth between 15 and 30 meters in October (d'Ortenzio et al., 2005), the month of largest occurrence of MTLC).



## 4 Discussion and Conclusion

This study provides a comprehensive analysis of Mediterranean Tropical-Like Cyclones (MTLC) and Extratropical Cyclones (ETC), based on a large dataset of real cyclones rather than individual case studies or model simulations. By leveraging this extensive dataset, we capture the diverse characteristics and developmental pathways of these cyclones, establishing a robust foundation for identifying precursors of the transitioning of classical extratropical perturbations into tropical-like cyclones in the Mediterranean region.

Both MTLC and intense ETC originate from a strong positive upper-level potential vorticity (PV) anomaly, setting them apart from the majority of ETC. Those strong PV anomalies are mainly of adiabatic origin, typically associated with PV advection from the stratosphere (Dolores-Tesillos et al., 2022; Sanchez et al., 2023)). While MTLC and intense ETC exhibit similar surface wind speeds, MTLC are associated with larger and more intense precipitation patterns, as well as stronger enthalpy fluxes at the air-sea interface, resulting in more dangerous phenomena. The primary distinction between these two
cyclone types lies in their potential intensity (PI) during the intensification phase. In this, seasonality plays a role: MTLC predominantly form in autumn and spring, whereas intense ETC develop in winter. Therefore, intense ETC are typically associated with colder sea surface temperatures (SST) than MTLC, conditions that are associated with lower climatological values of PI. In contrast, MTLC form at locations and times in which climatological PI is larger. Moreover, MTLC develop in situations when the actual PI is much larger than the climatological value, indicating that specific conditions localized in
time favor their occurrence. Wind shear further distinguishes the two cyclone types: MTLC form in environments with weaker upper-level jets, as highlighted in previous studies (Cavicchia et al., 2014a; Tous and Romero, 2013).

We have demonstrated that MTLC develop in presence of cold tropospheric anomalies (with respect to the climatological values), and that they typically occur over positive SST anomalies. This situation leads to a larger hurricane CAPE (with the tropospheric anomaly playing a bigger role than the SST anomaly), likely favoring the transition of the baroclinic disturbance
into a tropical-like cyclone.

The development of a cold anomaly in the mid-troposphere, driven by a strong upper-troposphere PV intrusion, can be explained by several dynamic and thermodynamic processes. A strong PV anomaly induces vertical motion below (Chaboureau et al., 2012), causing mid-tropospheric air to ascend and cool adiabatically as it expands under decreasing pressure. Additionally, the PV anomaly deforms isentropic surfaces, lowering them in the mid-troposphere and displacing colder, denser air
downward. The upper-level divergence associated with the anomaly reinforces this upward motion and cooling. Combined, these processes can generate a significant mid-tropospheric cold anomaly, destabilizing the air column in warm environments and enhancing convection. Further investigation of the terms in the omega equation, similar to the analysis of Hurricane Ophelia's extratropical transition by Rantanen et al. (2020), is needed to confirm the action of these processes.

Whereas it is relatively easy to interpret the cold tropospheric anomaly as a consequence of the low-pressure perturbation
itself, the warm SST anomaly associated with MTLC might be related both to a local or a larger-scale upper sea condition. This has not been investigated in this study.



In any case, while potential intensity (PI) in the Mediterranean region is lower than in tropical areas and typically not conducive to tropical transition, transient and/or localized increases in PI —driven by warm sea surface temperatures and strong potential vorticity anomalies that destabilize the atmosphere— can create conditions favorable for the wind-induced
surface heat exchange (WISHE)-like feedback. Under these conditions, MTLCs intensify through a combination of baroclinic and diabatic processes that enhance local PI, allowing WISHE to contribute to their development. These findings align with previous studies (Miglietta and Rotunno, 2019; Flaounas et al., 2021), though our analysis is based on reanalysis data rather than simulations.

The MTLCs in this dataset can also be considered within the framework proposed by Emanuel et al. (2024), which cate-
gorizes various extratropical cyclones —including polar lows, subtropical cyclones, Kona storms, and MTLCs— as systems driven by surface heat fluxes. In their paper, the authors propose the use of the name *cyclops* for cyclones with large PI associated with localized self-induced positive PI anomalies rather than due to a larger scale high PI environment, which is typical of the tropics and at times is found in the presence of particularly high SST in the Mediterranean regions (Emanuel et al., 2024).

Within this framework, we notice that our categorization of MTLC does not allow for a further differentiation between real
tropical cyclones, cyclops, and warm seclusions. Whereas we notice that the composite PI anomaly of MTLC is predominantly associated with a perturbation-induced cold tropospheric profile (which highlights the presence of cyclops in the MTLC group), a detailed study of PI and the relative contributions from cold troposphere and warm SST should be performed on individual cyclones to assess the frequency of cyclops.

However, Emanuel et al. (2024) attribute the collapse of these cyclones to a reduction in air column disequilibrium caused
by increased enthalpy in the mid-troposphere. In contrast, this study finds that MTLC decay is not linked to mid-tropospheric warming. Instead, a drop in SST appears to drive the PI decrease over time, resembling the SST feedback mechanism known in tropical cyclones, where surface cooling induced by the storm reduces PI (Schade and Emanuel, 1999; Jullien et al., 2014; Mei et al., 2012). However, ERA5 SST (which has been obtained using both spatial and temporal smoothing on observed SSTs) might not be the best available estimate of the actual sea surface temperature fields at the time of MTLC occurrence. The use
of purely observational SST datasets could lead to interesting findings in the response of the upper waters to MTLC, in the quantification of cold wakes associated with their passage, and in the investigation of their effect on the subsequent evolution of cyclones.

While this study benefits from a large observational dataset assimilated in ERA5 reanalysis, it is important to recognize the limitations of using for the investigation also non-assimilated variables, which may influence the interpretation of key cyclone
characteristics. ERA5 tends to underrepresent specific humidity in the lower and mid-troposphere, particularly in convective environments and over sea, which can bias estimates of thermodynamic disequilibrium and moisture fluxes (Johnston et al., 2021; Slocum et al., 2022; Pantillon et al., 2024). Biases in atmospheric temperature and humidity can distort the vertical instability structure and affect potential intensity metrics. Moreover, ERA5 systematically underestimates peak wind speeds in intense cyclones due to its relatively coarse resolution and smoothed surface drag parameterization, resulting in weaker near-
surface winds than those observed in both tropical and tropical-like systems (Gandoin and Garza, 2024; Gutiérrez-Fernández et al., 2023).



These shortcomings and the relatively coarse resolution of the used atmospheric model limit ERA5's ability to fully capture the mesoscale dynamics essential for cyclone development and intensification. This limitation is particularly evident when attempting to depict the most intense features—such as those found in the northwest quadrant of MTLC composites for both PI
and precipitation—where ERA5's spatial and physical resolution proves insufficient to resolve fine-scale structures. Therefore, future research should integrate high-resolution simulations (following Ragone et al. (2018); Bouin and Lebeaupin Brossier (2020); Sanchez et al. (2023)), fine scale hindcasts of reanalysis data (Bernini et al., 2025, such as the CHAPTER dataset), and targeted observations (like recommended by Velden et al. (2025)) to better resolve the fine-scale thermodynamic and dynamical processes governing Mediterranean tropical-like cyclones.

Finally, a key question opened by this study is whether potential intensity (PI) can serve as an early indicator of MTLC formation and intensification. This would involve identifying cyclones that exceed a PI threshold in their early stages and tracking the presence of a warm core throughout their lifecycle. Such an approach could improve MTLC prediction, enhancing preparedness and risk mitigation in the Mediterranean region.

*Data availability.* Hersbach et al. (2020) was downloaded from the Copernicus Climate Change Service (2023)

**Appendix**

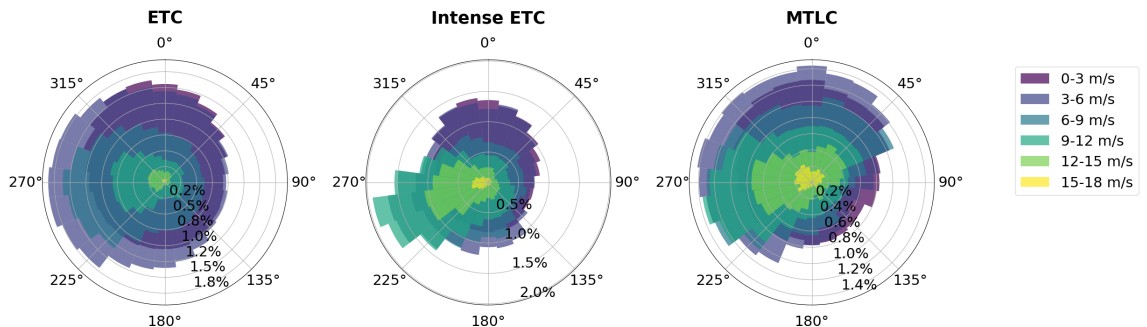

**Figure A1.** Wind rose at time of maximum cyclones' intensity for typical ETCs (left), intense ETCs (middle), and MTLCs (right).




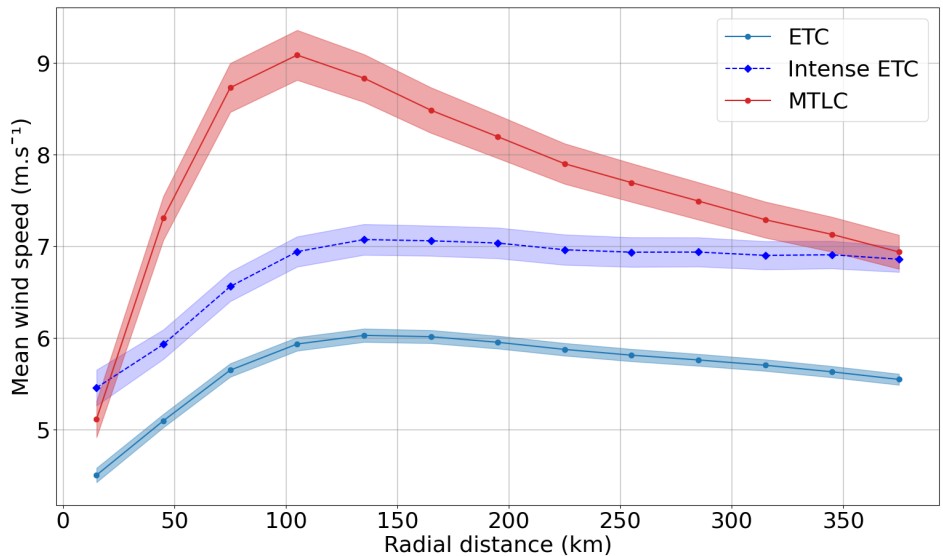

**Figure A2.** Azimuthal wind at time of maximum cyclones' intensity for typical ETCs (light blue solid line), intense ETCs (dark blue dashed line), and MTLCs (red solid line). Shading around the solid line indicates the standard error of the mean.

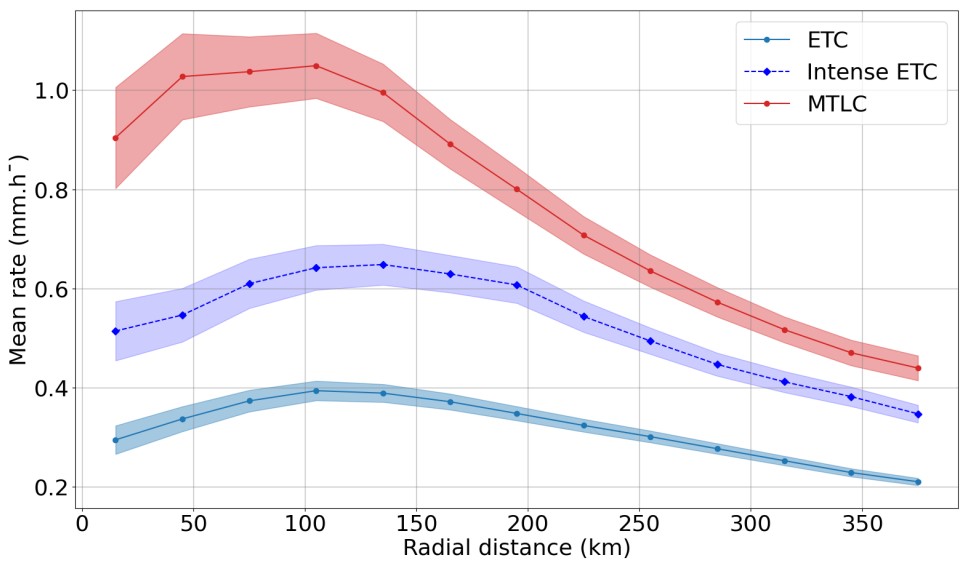

**Figure A3.** Azimuthal precipitation at time of maximum cyclones' intensity for typical ETC (light blue solid line), intense ETC (dark blue dashed line), and MTLC (red solid line). Shading around the solid line indicates the standard error of the mean.





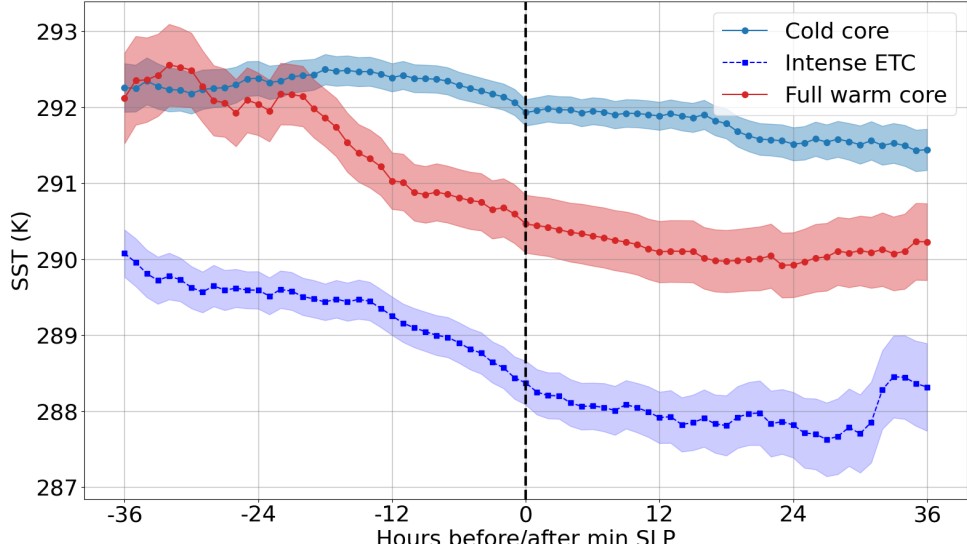

**Figure A4.** Composite time evolution of the mean SST for typical ETC (light blue solid line), intense ETC (dark blue dashed line), and MTLC (red solid line) with respect to their minimum SLP (black dashed line). The mean has been computed in a 10° by 10° box centered on each cyclone. Shading around the solid line indicates the standard error of the mean.



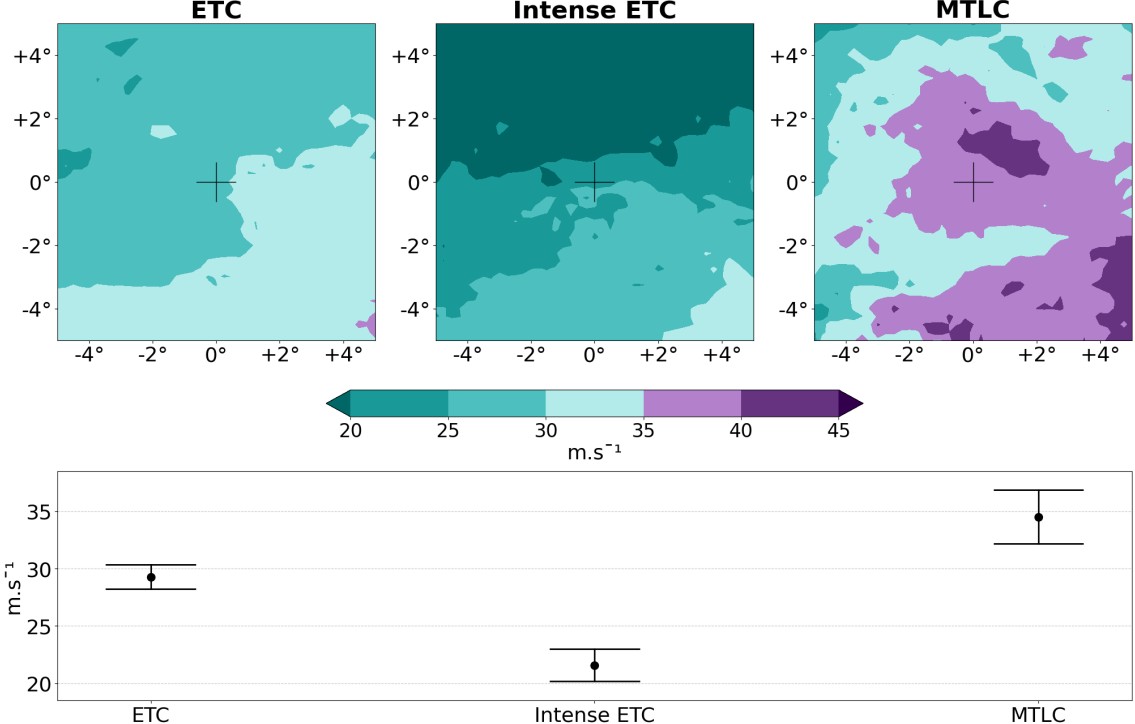

**Figure A5.** Climatological PI for the different cyclone classes. Top: 2D composites centered on the cyclones, 36h before the time of minimum SLP. The purple colors represent values of PI above $35\ m\ s^{-1}$, value below which tropical cyclones typically don't form (Emanuel, 2010). Bottom: box average of the climaatological PI shown in the maps above; error bars indicate the standard error of the mean over the different cyclones in each class.





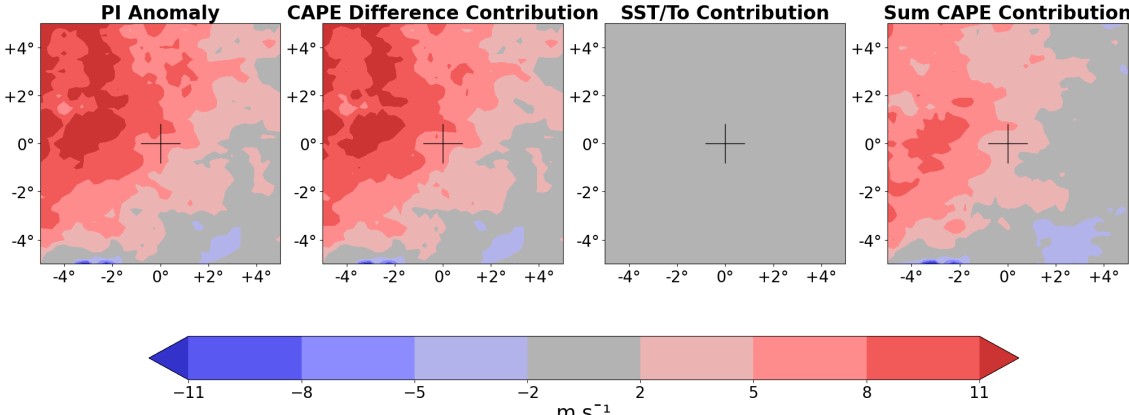

**Figure A6.** 2D composites of PI anomalies ($m\ s^{-1}$) for MTLC 36h before their minimum SLP. From the left: PI anomaly as the difference between the actual PI (**fig. 10**) and the climatological PI (**sup. fig. A5**); difference between the PI computed with all climatological values but the actual difference in CAPE and the climatological PI ; difference between the PI computed with all climatological values but the actual efficiency factor ($\frac{SST}{T_o}$) and the climatological PI; sum of the contributions to the anomaly created by the CAPE difference presented in **fig. 11**. The difference between the second and the fourth subplots shows the non-linearities at play in the total contribution of the CAPE difference to the final PI anomaly. The grey region, bounded by anomalies of ($4\ m\ s^{-1}$), has a width corresponding to twice the mean standard error of the climatological PI in MTLC.



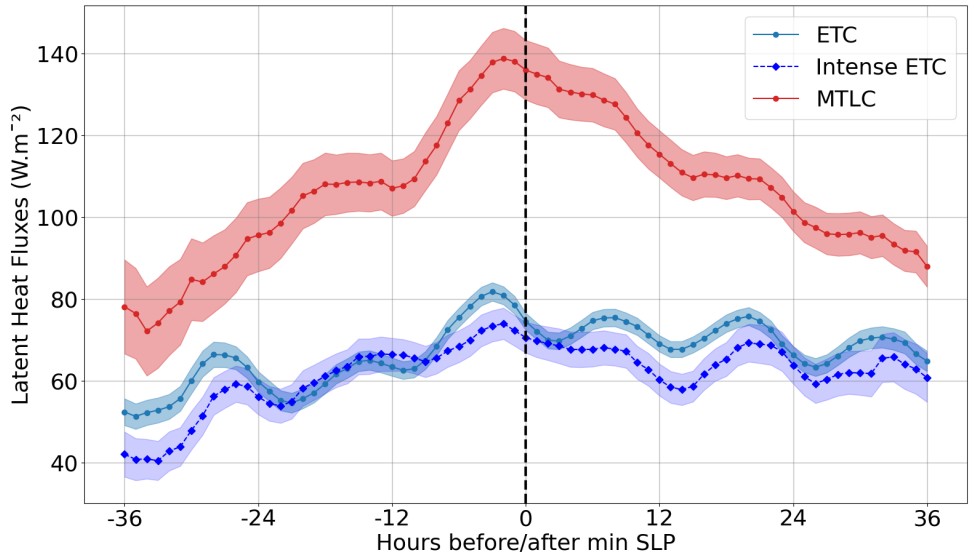

**Figure A7.** Composite time evolution of the mean air-sea latent heat flux for typical ETC (light blue solid line), intense ETC (dark blue dashed line), and MTLC (red solid line) with respect to their minimum SLP (black dashed line) . The mean has been computed in a 10° by 10° box centered on each cyclone. Only the points over the sea are considered. Shading around the solid line indicates the standard error of the mean.




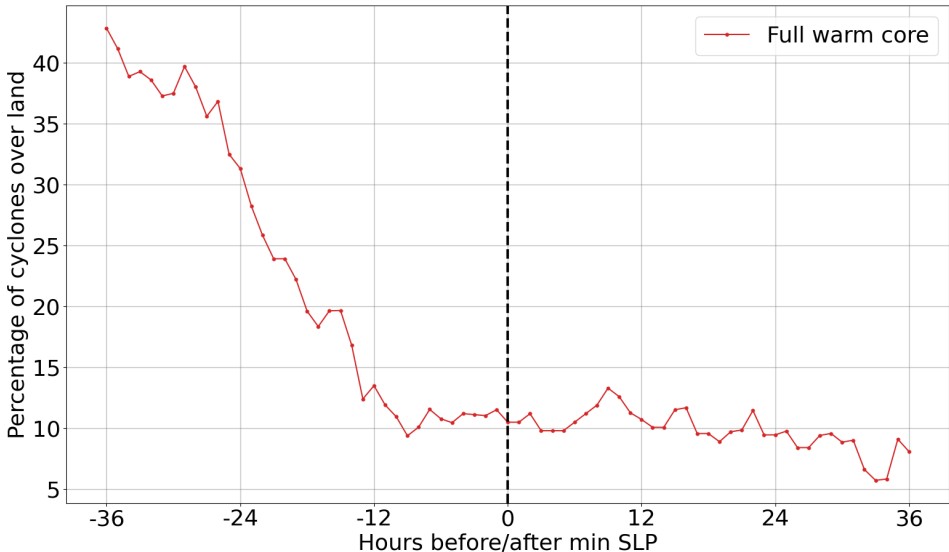

**Figure A8.** Time evolution of the percentage of cyclones' over land with respect to minimum SLP (black dashed line).

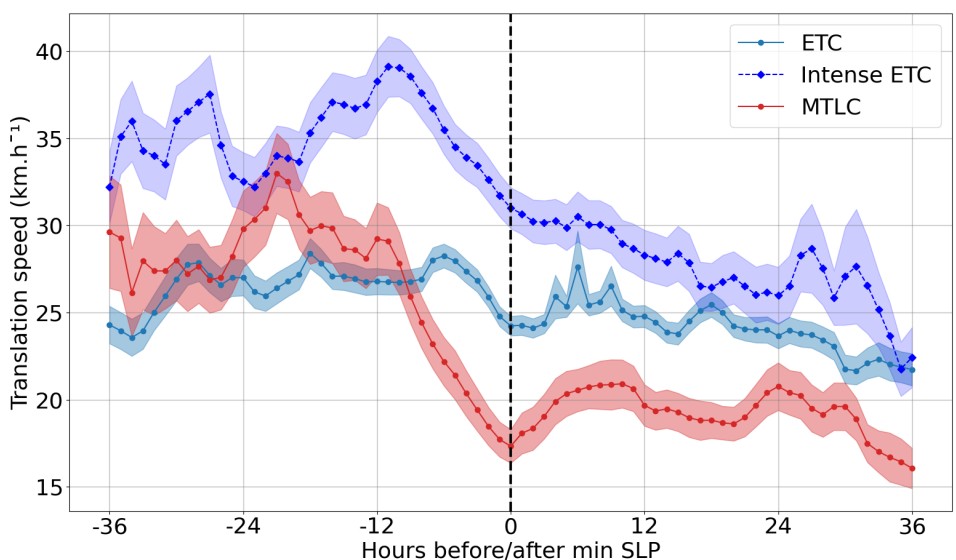

**Figure A9.** Composite time evolution of the translation speed for typical ETC (light blue solid line), intense ETC (dark blue dashed line), and MTLC (red solid line) with respect to their minimum SLP (black dashed line). Shading around the solid line indicates the standard error of the mean.



*Author contributions.* Conceptualization: L.B., L.C., and C.P.; methodology: L.B., L.C., E.S., F.D., and C.P.; formal analysis: L.B.; data curation: L.B.; computational resources: F. D. and C.P.; project supervision and administration: C.P.; original draft preparation: L.B. and C.P. All authors were involved in discussing the results and reviewing the manuscript.

*Competing interests.* The authors declare no conflict of interest.

*Acknowledgements.* This study is based upon work from COST Action MedCyclones (CA19109), supported by COST (European Cooperation in Science and Technology, https://www.cost.eu, last access: 01st August 2025). This work was also made possible by the TROPICANA program of the Institut Pascal at Université Paris-Saclay with the support of the program "Investissements d'avenir" ANR-11-IDEX-0003-01. We particularly thank Dr. E. Flaounas, Dr. M. M. Miglietta, and Dr. A. Parodi for their precious support through various and stimulating exchanges. Additionally, the research activities described in this paper have been partially funded by the Italian program "Piano Nazionale

di Ripresa e Resilienza – PNRR", Missione 4 Componente 2, Investimento 3.3 - D.M.352 09/04/2022, - M.U.R., Ministry of University and Research.



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
