# Peer review of "Environmental Characteristics Associated with the Tropical Transition of Mediterranean Cyclones"

_EGUsphere, 2025_

## Referee Comment (RC2)

**Review of *Environmental Characteristics Associated with the Tropical Transition of Mediterranean Cyclones**

By Bernini et al., submitted to *Weather and Climate Dynamics*

In this paper, the authors highlight the large-scale environmental conditions that are specific to the formation of warm-core Mediterranean Tropical-Like Cyclones (MTLCs) compared to the more typical cold-core Extra-Tropical Cyclones (ETCs) that form over the same basin. To do so, they composite the structure and environment of ETCs, intense ETCs and MLTCs from ERA5. This paper offers a good and comprehensive analysis of environmental factors related to MTLCs in the Mediterranean, utilizing a systematic climatological analysis that has been largely lacking in the Mediterranean Cyclones literature to date. As such, this is an important paper that fits within the scope of WCD. Should my main concerns be addressed, I would recommend it for publication.

**Major Comments**

1. My main concern is that composites, on which the paper relies for a significant portion, can be challenging and misleading, especially given the large number of samples. Indeed, a large number of samples are at risk of blurring out information when averaged. Here are suggestions to improve the robustness of the analysis:
   - Highlight significant areas, e.g. areas where the difference in a given variable between Intense ETCs and MTLCs is significant, or areas where an anomaly is significantly different from zero.
   - Provide composites of land-sea mask (in SI) to give an idea of how much land is included in these snapshots.
   - Does it make a difference if you orient the snapshots along the direction of propagation?
   - The 10° box is quite large for the Mediterranean, where it corresponds to the whole width of the basin, and the cyclones usually span a few hundreds of kilometers. Could you justify this choice? While averaging over a large area for environmental factors makes sense, I don't think it does for characterizing the cyclone itself (in particular when defining intensity).
2. Emanuel et al. 2025 used a modified version of the PI for identifying CYCLOPs, justifying that it was better suited for similar cases. Why not use it?

3. While I understand the CPS is the most commonly accepted way to classify MTLCs at the moment, you need to acknowledge ongoing debates in the Mediterranean Cyclones community and beyond regarding the limitations of the CPS.
4. Would your results change if you used wind instead of pressure for classifying he intensity?

**Minor Comments**

Most of these do not require a specific answer. I am highlighting in bold and with a star those I would like an answer to.

5. L. 28 you write "their cyclogenesis is different", however, some TCs in the Atlantic also form from Tropical Transition (see McTaggart-Cowan et al.).
6. L. 37 for clarity of the structure, I would not start a new paragraph here.
7. L. 66 "first occurrence" -> "the first time this minimum is reached": more clear.
8. L. 67 and several afterwards: the word "despite" is misused. In this case, replace with "even though"? Check and fix all following occurences (L. 186).
9. L. 96, 126, and several times afterwards: ETCs should take an s when plural.
10. **\*L. 96-98: Can you explain your choice to add a tolerance for $-V_u$ only? Why not $-V_l$? And why six hours?**
11. L. 106-107: This is a very long list of references, most of which are unrelevant to the present work. I would advise to cut it down with only a few that are closest to the present work and adding "e.g.".
12. L 124: ETC is Extra-Tropical Cyclone, so "ETC Cyclones" is a repetition.
13. L. 127: "These represent 192" or "There are 192 such/intense cyclones". Otherwise unclear.
14. L. 130-138: This remark is very relevant and appreciated.
15. **\*L. 141: How long does the warm core itself last for?**
16. L. 146: Precise "(not shown)".
17. Figure 1:
    a. **\*For the "intense ETCs" lines : the fact that numbers drop sharply on each side on the peak suggest to me that a small but significant portion of them have their maximum intensity at the very beginning or very end of their track. This might not be desirable. Can you investigate this?**
18. L. 158: I think it should be said earlier that you include the Black Sea in your analysis, as it is not necessarily obvious that you would. Also when comparing your frequencies to other papers, caution must be taken whether they included it or not.

19. **\*L. 166: I would expect the maximum in intense ETCs to be in winter. Can you explain why it is not the case? Please compare to other references with similar analyses.**

20. Figure 3:
    a. I would recommend superimposing the three plots into one panel for easier comparison.
    b. Number seem low compared to the total frequency, especially for ETCs: You announce 23 cyclones per year, but the sum of all the bars is far from it. Can you explain this discrepancy?
    c. Do you count the cyclones at their genesis, maximum intensity, or something else?

21. Figure 4:
    a. The choice of colors for temperature is unsettling. Can you use a blue-to-red colormap?
    b. You need to explain what the bottom panel is in the caption. It is also true for all following figures with this panel.

22. L. 182: Discuss limitations associated with ERA5's precipitation. Potentially (but I understand this is more work and might be out of your scope), you may consider retrieving precipitations from other sources, e.g. IMERG.

23. **\*L. 196: This is a very large box compared to the typical cyclone size, especially over the Mediterranean sea, where cyclones' size are usually a few hundreds of kilometers, and given 10° is basically the width of the whole basin. Taking the average is also disputable since you may average more or less lad depending on the position of the storm. Can you explain these choice? A usual choice is to take the maximum within 2° of the cyclone's center. Why not do this? I would recommend using a box no larger than 5° in any case, and also preliminarily masking all winds over lands, as they may be spuriously high due to orography, or overall low due to friction.**

24. L. 210: This could be discussed in light of the significance tests I suggest above.

25. L. 213: "indicating" -> "suggesting" seems a better level of confidence given your evidence.

26. L. 216-... : Please rewrite the start of this paragraph, which is unclear. I only understood what you are doing after reading the corresponding results.

27. L. 224-229: Did you se the CAPE as provided in the tcpypi package, or did you compute it separately. In my experience, tcpypi's CAPE does not equal the CAPE computed by other packages e.g. MetPy, so if you used different packagesit might lead to discrepancies.

28. L. 238-... : The insertion of figure 12 in this paragraph is confusing. I would suggest restricting this paragraph to what you read from figure 11, and then adding what you learn from figure 12 later.

29. In fact, figure 12 does not seem to be discussed anywhere else? If you do not wish to, it should be moved to the SI.

30. L. 270 Did you mean "In the case of hurricanes, the reduction in CAPE leads to lower PI" and you investigate whether this is the case for MTLCs?

31. L. 276-278: It is not clear to me what role the shallow mixed layer plays in this case, could you detail slightly more?

32. L. 287: Extra ')' after citation.

33. L. 291: "Intense ETCs develop in winter": You said they developed predominantly in spring in the corresponding paragraph.

34. L. 298: What do you mean "hurricane CAPE"?

---

## Editor Comment (EC1)

The manuscript entitled "Environmental characteristics associated with the development of tropical-like features in Mediterranean Cyclones" makes use of **ERA5** and the **Flaounas et al. Mediterranean cyclone track catalogue** to identify all cyclones from 1979–2020 and then classifies them using the **Hart cyclone phase space** into:

- ETC: cold-core extratropical cyclones
- DWCC: "deep warm-core cyclones" (their broader umbrella including medicanes / MTLCs)
- WETC / SETC: weak vs **strong** ETC (15 % deepest SLP)

They then compare environments and evolution of DWCC vs SETC and WETC:

- SST, climatological and instantaneous **Potential Intensity** (PI) and its decomposition into contributions from SST vs tropospheric temperature / humidity
- Upper-level PV anomalies and vertical wind shear
- Winds, precipitation, surface fluxes, and SST cooling along the track to discuss **intensification and decay mechanisms**

I don't see any fundamental scientific error or contradiction with the existing medicane / MTLC literature. The contribution to the current Mediterranean cyclone literature though can be characterized as incremental: it quantifies, on a relatively large sample, how potential intensity, SST, wind shear and PV anomalies differ between deep warm-core Mediterranean cyclones and other strong Mediterranean cyclones, and it links decay to cold-wake feedback in a way that's consistent with tropical-cyclone theory.

Main messages:

- DWCC and strong ETC reach **similar peak winds**, but DWCC produce stronger precipitation and stronger surface fluxes.
- 36 h before peak, DWCC tend to be over **warmer SST**, **higher PI**, and **lower vertical wind shear** than strong ETC, while **PV anomalies are similar** between the two groups.
- The **PI anomalies** that distinguish DWCC are mainly due to an **anomalously cold mid-troposphere** (linked to upper PV intrusion) plus a more modest SST anomaly; lower-tropospheric humidity and temperature anomalies are secondary.
- During the decay, DWCC experience **~3 K SST cooling** and a corresponding PI reduction; sensitivity tests with "only SST evolving" PI suggest the SST drop is the main contributor to the PI decrease, consistent with cold-wake feedback known from tropical cyclones.

They explicitly connect their results to:

- Classic medicane climatologies and environmental studies (e.g. Cavicchia et al. 2014; Ragone et al. 2018; Miglietta & Rotunno 2019)
- The new **CYCLOPs** framework and modified PI concept of Emanuel et al. (2024/2025).
- The 2024 GRL paper "A New Refinement of Mediterranean Tropical-Like Cyclones Characteristics", which also uses ERA5 and CPS but focuses on cyclone *classification* rather than environmental PI decomposition.

**What this paper adds to our understanding:**

1. **Systematic comparison with strong cold-core ETC in the same basin.** Rather than studying MTLCs in isolation, they explicitly compare DWCC to the subset of "strong" ETC with similar intensity (in SLP and winds) and similar upper-level PV anomaly, and show that what *distinguishes* DWCC is essentially:
   - higher PI and warmer SST,
   - weaker shear,
   - and stronger precipitation at peak.
2. **Decomposition of PI anomalies for a large sample.**

   There have been case studies and theoretical arguments that upper-level PV intrusions and SST both modulate PI, but here they quantify for many events that the dominant contribution to PI anomalies near DWCC comes from a **cold mid-tropospheric temperature anomaly**, with a secondary contribution from SST, and very minor roles for near-surface humidity/temperature.

   That's a meaningful clarification of *how* PI is raised in these events and links nicely to the CYCLOPs picture.

3. **Climatological cold-wake / decay discussion.**

   Cold-wake feedback on medicane intensity has been discussed mostly in case studies or coupled simulations. This paper makes a first attempt at a **climatological composite** showing that DWCC experience large (~3 K) cold wakes, and that PI decay in composites is mostly explained by SST evolution, reinforced by slower translation speed and shallow mixed layers in the Mediterranean.

Overall, I'd characterise the novelty as **moderate**. It does not contradict the literature, but it mostly tightens and quantifies patterns that had been hypothesised or shown in small samples. I would recommend some major revisions before recommending it for publication.

**Major scientific and conceptual issues**

1. The paper introduces **DWCC** as a thermal-structure class based purely on CPS warm core (−Vl, −Vu) over sea for ≥6 h, then often speaks about "tropical-like cyclones" in a way that suggests DWCC ≈ MTLC/medicane.

   But CPS warm cores in ERA5 at this resolution can include **warm seclusions and hybrids** that many in the community would not call "medicanes" in the strict sense (highly axisymmetric, deep convection around an eye). The recent GRL paper by Gutiérrez-Fernández et al. explicitly argues for refining that distinction.

   Suggestion: Make the taxonomy very explicit up front: DWCC is a CPS-based class that includes medicanes, CYCLOPs, and some warm seclusions, *not a 1:1 medicane list*. Also, be consistent in using "DWCC" for results, and reserve "medicane/MTLC" for when they refer to the historical literature or clearly defined subsets. Finally, add a short quantitative estimate of how many DWCC correspond to known medicane events or to stricter MTLC definitions, if possible.

2. **Dependence on CPS and ERA5 resolution:** The authors do acknowledge the limitations of CPS and ERA5 (resolution, humidity and wind biases, structure of convective systems). But in the actual interpretation, those limitations are sometimes underplayed.bIssues to highlight in the text:

   **CPS limitations:** Only −Vl and −Vu are used (no B), which is sensible given known issues with asymmetry, but that accentuates the risk of misclassifying warm seclusions as "tropical-like". The authors should be even more explicit about what kinds of storms might be included in DWCC and how that might bias conclusions (e.g. any tendency to favour systems near strong baroclinic zones).

3. **Causality vs correlation in the decay phase:** The authors argue in Lines 424-426 that DWCC decay is "typically linked to a drop in SST, which drives the decrease of PI over time", aligning with TC cold-wake feedback theory.
   But what they show is:

- A composite PI time series that decreases from −36 to +36 h.
- A composite SST time series and a PI sensitivity experiment in which letting only SST vary reproduces most of the PI decrease.

- A composite drop in translation speed that would itself enhance cooling and exposure to cooled water.

This is **strong circumstantial evidence but not a direct causal demonstration**.

I suggest to soften statements from "is linked to / drives" to "our composites are consistent with the SST-controlled PI feedback known in TCs" and be explicit that other factors (changes in shear, upper-level PV environment, baroclinic forcing) are not ruled out.

And/or: If feasible within the scope, they could add a simple scatterplot / correlation across individual DWCC between:

- magnitude of SST cooling vs change in PI vs change in intensity,
- controlling for translation speed and initial PI.

**4. How different are DWCC and SETC?**

From the response to reviewers and the boxplots, it's clear that:

In many metrics (SST, PI, shear, PV, precipitation), the **distributions of DWCC and SETC overlap strongly**, with statistically significant differences in means but large intra-class spread.

The paper currently emphasises the mean differences but does not make it crystal clear to a non-expert reader that: These variables are *not* sufficient to cleanly classify an individual cyclone as DWCC vs SETC; they only indicate tendencies that favour DWCC development.

I would suggest to adde near the end of Section 3.2–3.3, a short paragraph highlighting this, for example:
*"Despite statistically significant differences in mean SST, PI and shear between DWCC and SETC, the spread and overlap in the distributions is large (see boxplots). None of these environmental parameters alone can be used as a reliable classifier; instead, they highlight the conditions that favour the development of deep warm cores."*

5. Make a better relationship to CYCLOPs and modified PI

Even though the authors cite Emanuel 2024/2025 and the modified PI definition, they explicitly argue that many DWCC in their sample are probably "cyclops" in the Emanuel sense (PI anomalies mostly due to tropospheric cooling rather than basin-wide high SST), and Emanuel et al. motivate a modified PI metric precisely for such systems.

The choice of the classic PI instead of the modified one should be better explained.

**General comments and space for presentation improvement and clarity:**

- **Introduction:** In Lines 59-63, I see the main motivation to conduct such a study, but is this what you really show? From my understanding, your results show how the environments of deep warm-core Mediterranean cyclones differ from those of other strong Mediterranean cyclones. You also show some thermodynamic factors that contribute most to raising PI in DWCC and investigate the cold wake and its influence during the DWCC decay.

- Check consistency of terminology: "medicane", "MTLC", "tropical-like cyclone", "DWCC". Right now they sometimes seem interchangeable.

- In the decay section, explicitly mention whether landfalling DWCC (the ~25 %) have a distinct PI/SST evolution compared to fully maritime ones; even a one-sentence statement that composites exclude or include landfalling systems would clarify.

---

## Author Comment (AC1)

[Figure]

Figure R1: Probability density function of the sea level pressure associated with the center of ETC (blue) and MTLC (red) at the time of their maximum intensity.

[Figure]

Figure R2: Probability density function of the track length of ETC (blue) and MTLC (red).

[Figure]

Figure R3: Time evolution with respect to minimum SLP of the number of intense ETC (blue solid line) and MTLC (red solid line), proportion of MTLC that have a warm-core at each time step (maroon dashed line), and number of MTLC that have a deep warm core for the first time (dotted orange line).

[Figure]

Figure R4: Sea Surface Temperature for the different cyclone classes. Top: 2D composites centered on the cyclones, 36h before the time of minimum SLP. Bottom: boxplots of the mean SST for each cyclone, where the mean is computed on the 10% highest values over the 10°x10° box for each cyclone. Whiskers show the 5th–95th percentiles.

[Figure]

Figure R5: 10m surface wind for the different cyclone classes. Top: 2D composites centered on the cyclones, at the time of minimum SLP. Bottom: boxplots of the mean 10-m wind speed for each cyclone, where the mean is computed on the 10% highest values over the 10°x10° box for each cyclone. Whiskers show the 5th–95th percentiles.

[Figure]

Figure R6: Hourly precipitation for the different cyclone classes. Top: 2D composites centered on the cyclones, at the time of minimum SLP. Bottom: boxplots of the mean

1-h accumulated precipitation for each cyclone, where the mean is computed on the 10% highest values over the 10°x10° box for each cyclone. Whiskers show the 5th–95th percentiles.

[Figure]

Figure R7: PV anomaly for the different cyclone classes. Top: 2D composites centered on the cyclones, 36h before the time of minimum SLP. Bottom: boxplots of the mean PV anomaly for each cyclone, where the mean is computed on the 10% highest values over the 10°x10° box for each cyclone. Whiskers show the 5th–95th percentiles.

[Figure]

Figure R8: Total PI for the different cyclone classes. Top: 2D composites centered on the cyclones, 36h before the time of minimum SLP. Bottom: boxplots of the mean PI for each cyclone, where the mean is computed on the 10% highest values over the 10°x10° box for each cyclone. Whiskers show the 5th–95th percentiles.

[Figure]

Figure R9: Climatological PI for the different cyclone classes. Top: 2D composites centered on the cyclones, 36h before the time of minimum SLP. Bottom: boxplots of the mean climatological PI for each cyclone, where the mean is computed on the 10% highest values over the 10°x10° box for each cyclone. Whiskers show the 5th–95th percentiles.

[Figure]

Figure R10: Wind shear for the different cyclone classes. Top: 2D composites centered on the cyclones, 36h before the time of minimum SLP. Bottom: boxplots of the mean wind shear for each cyclone, where the mean is computed on the 10% highest values over the 10°x10° box for each cyclone. Whiskers show the 5th–95th percentiles.